# Vanadium-Substituted Dawson-Type Polyoxometalate–TiO_2_ Nanowire Composite Film as Advanced Cathode Material for Bifunctional Electrochromic Energy-Storage Devices

**DOI:** 10.3390/molecules27134291

**Published:** 2022-07-04

**Authors:** Yu Fu, Yanyan Yang, Dongxue Chu, Zefeng Liu, Lili Zhou, Xiaoyang Yu, Xiaoshu Qu

**Affiliations:** College of Chemical and Pharmaceutical Engineering, Jilin Institute of Chemical Technology, Jilin 132022, China; fuyu1996211@163.com (Y.F.); chudongxue1997@163.com (D.C.); lzf122321@163.com (Z.L.); z13069189739@163.com (L.Z.); yangyangyu@jlict.edu.cn (X.Y.)

**Keywords:** polyoxometalate, TiO_2_ nanowire, composite film, bifunctional electrochromic energy storage

## Abstract

Polyoxometalates (POMs) demonstrate potential for application in the development of integrated smart energy devices based on bifunctional electrochromic (EC) optical modulation and electrochemical energy storage. Herein, a nanocomposite thin film composed of a vanadium-substituted Dawson-type POM, i.e., K_7_[P_2_W_17_VO_62_]·18H_2_O, and TiO_2_ nanowires were constructed via the combination of hydrothermal and layer-by-layer self-assembly methods. Through scanning electron microscopy and energy-dispersive spectroscopy characterisations, it was found that the TiO_2_ nanowire substrate acts as a skeleton to adsorb POM nanoparticles, thereby avoiding the aggregation or stacking of POM particles. The unique three-dimensional core−shell structures of these nanocomposites with high specific surface areas increases the number of active sites during the reaction process and shortens the ion diffusion pathway, thereby improving the electrochemical activities and electrical conductivities. Compared with pure POM thin films, the composite films showed improved EC properties with a significant optical contrast (38.32% at 580 nm), a short response time (1.65 and 1.64 s for colouring and bleaching, respectively), an excellent colouration efficiency (116.5 cm^2^ C^−1^), and satisfactory energy-storage properties (volumetric capacitance = 297.1 F cm^−3^ at 0.2 mA cm^−2^). Finally, a solid-state electrochromic energy-storage (EES) device was fabricated using the composite film as the cathode. After charging, the constructed device was able to light up a single light-emitting diode for 20 s. These results highlight the promising features of POM-based EES devices and demonstrate their potential for use in a wide range of applications, such as smart windows, military camouflage, sensors, and intelligent systems.

## 1. Introduction

With the continuing development of sustainable resources, devices for energy storage and conversion, such as solar cells, supercapacitors, and electrochromic (EC) devices, have attracted increasing attention [1,2,3]. EC devices are known to change colour via charge insertion/extraction or reversible redox reactions driven by an external electric field [4,5]. Simultaneously, the ion intercalation/deintercalation steps taking place during the reversible redox reactions of the EC process can also generate a pseudocapacitive behaviour [6,7], thereby resulting in EC devices and supercapacitors having similar working mechanisms and device structures. Based on this principle, one can envisage that these two functions could be integrated into a single electrochromic energy-storage (EES) device using the same material. As such, several EES devices have been widely explored. For example, Feng et al. [8] utilised exfoliated graphene/V_2_O_5_ as the active material of a micro-supercapacitor to judge its charge-discharge state via the observed colour. In addition, Xue et al. [9] synthesised a smart EC supercapacitor device using a porous co-doped NiO film as the positive electrode. This device exhibited a high specific capacitance, high energy density, and good cycle stability. After charging, these two devices were able to light up light-emitting diodes (LEDs).

Among the various EES materials reported to date, polyoxometalates (POMs) demonstrate a multi-electron reaction specificity during the electrochemical redox process, which contributes to their chromatic transitions and high-efficiency energy-storage performances [10,11,12]. As an example, Ma et al. [13] synthesised a POMs-based supramolecular crystalline material, namely, H_3_PMo^VI^_12_O_40_·(BPE)_2.5_·3H_2_O (BPE  =  1,2-Bis(4-pyridyl)ethylene), via a one-step hydrothermal method. The compound had a high specific capacitance (i.e., 137.5 F g^−1^ at 2 A g^−1^) and good cycle stability (i.e., 92.0% after 1000 cycles) than parent H_3_PMo^VI^_12_O_40_. This work provided an alternative method for improving the performance of POMs-based capacitor electrode materials. In addition, Wang et al. [14] reported a high-performance PW_12_-based EC device, wherein the optical contrast of the optimised device containing an I^−^/I^3−^ redox couple in the electrolyte reached 59.4%. The PW_12_-EC device also showed a fast response time for bleaching and colouration. However, POM materials tend to aggregate or stack to form dense structures, which can hinder ion diffusion and affect their electrochemical properties. To overcome this issue, the incorporation of POMs into nanostructures or composite materials has been investigated to increase their surface areas [15]. For example, Li et al. [16] prepared graphene oxide/W_18_O_49_ nanorod (rGO-WNd) composites through the high-temperature thermal reduction of ammonium tungstate and graphene oxide (GO). Compared with the cycle stability, capacitance, and EC properties of the pure WNd film, the corresponding properties of the Rgo-WNd composite film were significantly enhanced. This could be attributed to a higher degree of ion diffusion and the acceleration of charge transfer after the addition of rGO. As a result, the response times of such materials are improved.

Titanium dioxide is recognised as a promising candidate for EC and energy-storage applications owing to its excellent electrochemical stability, optical modulation, reversibility, and mass transport properties, as well as the fact that it enhances contact with the electrolyte and improves the resulting reaction kinetics [17]. In recent years, various TiO_2_ nanostructures, such as nanorods, nanotubes, and nanowires, have received attention as excellent composite materials because of their large specific surface areas and orderly structures. For example, Khanna et al. [18] fabricated a TiO_2_@NiTi system for use as an electrode in energy-storage applications, and this material produced a specific capacitance of ~1 F g^−1^. This result reveals that their system is a promising material for energy-storage applications. In addition, Ji et al. [19] designed and fabricated a novel bilayer composite with an excellent energy-storage performance by combining an aligned TiO_2_ nanoarray (TNA) and random TiO_2_ nanowires (TiO_2_ NWs) with a poly(vinylidene fluoride) (PVDF) matrix. A superior discharge energy density of 16.13 J cm^−3^ was obtained for the 5 vol% TiO_2_ NW/TNA-PVDF composite, which was 2.0 times higher than that of the pure PVDF matrix (8.23 J cm^−3^). Furthermore, Lv et al. [20] synthesised TiO_2_ nanotube membrane electrodes that exhibited excellent EC performances, combining a high colouration contrast with a transmittance of 65% in the visible spectrum, in addition to a good cycle stability (88.2% for initial optical modulation after 1000 cycles). Zhang et al. [21] reported a novel EC device based on polyaniline nanofibers wrapped with antimony-doped tin oxide/TiO_2_ nanorods (ATO/TiO_2_@PANI film) as an EC electrode material. Compared with the pure PANI film, the EC device based on ATO/TiO_2_@PANI film shows better electrochromic performance.

Based on the above considerations, our group previously designed a series of POM-based EC thin film materials [22,23]. In 2020, we reported the first dual-function electrochromic-energy storage material based on POMs and TiO_2_ nanowires [24]. However, the response time of the film is long, and its capacitive performance is relatively low. As we know, the structure and composition of POMs have a great influence on their electrochemical activity; therefore, the electrochromic-energy storage properties could be adjusted easily by changing the type of POMs. In general, the lacunary and substituted Dawson structures can show enhanced electrochromic performances [25]. Thus, in the current study, to improve the performances of these materials, we chose vanadium-substituted Dawson-type polyoxotungstate K_7_[P_2_W_17_VO_62_]·18H_2_O (P_2_W_17_V) and TiO_2_ nanowires to fabricate a nanocomposite thin film via hydrothermal and layer-by-layer (LbL) self-assembly methods. The microstructure of TiO_2_ is regulated by a hydrothermal treatment, allowing its nanowire array to be employed as the substrate for the composite film. The synergistic effects of the TiO_2_ NWs and the POMs could improve the EC properties of the composite film. Scanning electron microscopy (SEM), transmission electron microscopy (TEM), atomic force microscopy (AFM), and X-ray photoelectron spectroscopy (XPS) are used to investigate the surface morphology, structure, and chemical properties of the obtained nanocomposite film. Finally, the EC and energy-storage properties of the composite film are compared with those of the pure P_2_W_17_V-modified fluorine-doped tin oxide (FTO) film.

## 2. Materials and Methods

### 2.1. Chemicals and Materials

All reagents were of analytical grade and used as received without further treatment. The FTO-coated glasses (<10 ohm sq^−1^) were purchased from Pilkington (Toledo, Ohio, USA). (3-Aminopropyl)trimethoxysilane (APS), polyetherimide (PEI), and propylene carbonate (PC) were purchased from the Aladdin Chemical Co., Ltd. and were used without further treatment. P_2_W_17_V was prepared according to a method reported in literature [26,27], and it was characterised by infrared (IR) spectroscopy (Appendix A), ultraviolet-visible (UV-vis) absorption spectroscopy (Appendix A), and cyclic voltammetry (CV) (Appendix A).

### 2.2. Preparation of the Composite Films

The TiO_2_ nanowire arrays were prepared via a hydrothermal synthesis method according to our previous report [22], and this was followed by the preparation of the composite films. More specifically, the surface of the FTO substrate was cleaned in NH_3_/H_2_O/H_2_O/H_2_O_2_ (volume ratio 1:1:1) at 80 °C for 20 min and then rinsed with deionised water. This step was repeated 3*–*5 times to remove any inorganic and organic impurities from the FTO substrate. The composite film was prepared via the LbL assembly method. Initially, the cleaned FTO substrate was modified with TiO_2_ NWs. Subsequently, the pure FTO and the modified FTO were immersed in APS overnight. After this time, the samples were placed in HCl (pH 2.0) for 20 min, rinsed with deionised water, and dried under a stream of nitrogen to give the precursor. Finally, the composite film (NW−P_2_W_17_V) was constructed by depositing negatively charged P_2_W_17_V (5 × 10^−3^ mol L^−1^ in 0.2 mol L^−1^ HOAc-NaAc at pH 3.99) and positively charged PEI (5 × 10^−3^ mol L^−1^ at pH = 4) onto the TiO_2_ NWs, according to the LbL method. For comparison, an additional film was prepared on the pure FTO substrate using the same method, and this was designated as FTO−P_2_W_17_V. A schematic outline of the fabrication process is shown in Figure 1a.

### 2.3. Characterisation

SEM images were measured on FEI Verious 460 L scanning electron microscope (Hillsboro, OH, USA). AFM images were investigated by Icon Bruker microscope (Ettlingen, Germany). TEM images were measured on a FEI Tecnai G2F20 S-TWIN microscope equipped with an energy-dispersive spectrometer (EDS) (Hillsboro, OH, USA). XPS analysis were measured on a Thermo ESCALAB 250 spectrometer (Shanghai, China). The EC and capacitive properties of the films were determined by combining the in-situ TU-1901 PERSEE UV-vis spectrophotometer (Beijing, China) with an CHI660B Chenhua electrochemical workstation (Shanghai, China) in a three-electrode configuration, where the nanocomposites served as the working electrodes; a Pt plate/Pt wire acted as the counter electrode, and Ag/AgCl was used as the reference electrode.

## 3. Results and Discussion

### 3.1. Characterisation of the NW−P_2_W_17_V and FTO−P_2_W_17_V Materials

The multilayer growth process of composite film on the precursor-coated quartz substrate (on both sides) was monitored by UV-Vis spectroscopy (Figure 1b). It exhibited strong absorption of P_2_W_17_V with two characteristic absorption peaks at 201 and 289 nm. The peak at 201 nm originates from the terminal oxygen to tungsten charge-transfer transition (O_d_→W), whereas the peak at 289 nm corresponds with the charge-transfer transition from the bridging-oxygen to tungsten (O_b_/O_c_→W). The inset of Figure 1b shows the plots of the absorbance values at 201 and 289 nm as a function of the layer number and suggests that growth is uniform during each cycle.

SEM-EDS and TEM were then performed to obtain the detailed information about the surface morphologies and homogeneities of the composite materials. The SEM images of the FTO−P_2_W_17_V film are shown in Appendix A, wherein it can be visualised that the FTO substrate was covered by aggregated P_2_W_17_V anions. In addition, the cross-sectional view of the FTO−P_2_W_17_V film gave a thickness of ~150 nm. As shown in Appendix A, the FTO substrate was covered with densely grown TiO_2_ NWs, and the cross-sectional image confirmed that the height of the nanowires was approximately 600 nm. After the LbL process, it was apparent that the interspaces of the NWs were filled, and the NWs became wider and more compact owing to the deposition of P_2_W_17_V and PEI (Figure 2a). Moreover, the EDS mapping of P, W, Ti, and V confirmed the feasibility of the hydrothermal treatment and LbL process (Figure 2b), since the POMs and the TiO_2_ NWs were evenly distributed on the surface of the FTO substrate.

Subsequently, AFM was employed to study the surface morphologies and roughness properties of the FTO−P_2_W_17_V and NW−P_2_W_17_V films (Figure 2c,d and Appendix A). Two-dimensional (2D) and three-dimensional (3D) images of the two films confirmed that their surface microstructures were quite different. More specifically, the AFM images of the FTO−P_2_W_17_V film displayed some uniformly sized spherical particles, which resulted from the FTO substrate being covered with cross-linked POM anions with a thickness of 100 nm (Appendix A). From Figure 2d, it was apparent that the surface of the NW−P_2_W_17_V film shows a regular cylindrical microstructure, suggesting the presence of TiO_2_ NWs substrate. The height of the NWs anchored with the POMs was ~500 nm, which corresponded well with the SEM observations. In addition, the root mean square (RMS) roughness for each film was calculated from an area of 5 × 5 μm^2^ in the AFM image, wherein the surface roughness (i.e., RMS) values of the NW−P_2_W_17_V and FTO−P_2_W_17_V films were found to be 73.6 and 20.5 nm, respectively. A higher roughness could lead to a larger reactive surface area, thereby improving the electrochemical performance of the material.

The surface chemical compositions of the as-prepared films were further determined and quantified by XPS analysis. The high-resolution XPS spectra of the prepared composite film shown in Figure 2 indicates that the composite material mainly contains C, P, Ti, and W [28,29], wherein the Ti should originate from the TiO_2_ NWs on the FTO substrate. This result further confirms that the POMs and the TiO_2_ NWs are distributed on the surface of the FTO substrate. As shown in Figure 2e, the most intense doublet peaks are observed at 35.6 and 37.7 eV, which correspond to the binding energies of the electrons in the W4f_7/2_ and W4f_5/2_ levels of W in the W^(VI)^ valence state. These results indicate that the majority of W atoms were in a highly oxidised state and could be reduced to W^(V)^, which is the key reaction in the EC process of polyoxotungstate-based materials. With respect to the high-resolution Ti2p peaks, they could be split into peaks at 458.9 and 464.6 eV, which were both attributed to TiO_2_ (Figure 2f), thereby indicating that the main matrix component was TiO_2_. Furthermore, the prepared film exhibited a peak corresponding to the C1s level (284.8 eV) of the carbon present in the PEI polycation, whereas the P2p signal (at 133.0 eV) and the V2p signal (at 532.4 eV) [30] were ascribed to P_2_W_17_V (Appendix A). Thus, the XPS data suggest that PEI cations and P_2_W_17_V anions were incorporated into the TiO_2_ NW substrate, which is consistent with the UV-vis results.

TEM is indispensable for the characterisation of nanostructured materials, particularly when the particle shape is important in determining its function, and so TEM was employed herein to evaluate the microstructure of the composite and the spatial relationship between TiO_2_ NWs and P_2_W_17_V. Figure 3a–b show the typical TEM images of the TiO_2_ NWs with a diameter of ~50 nm. The EDS elemental mapping patterns of the TiO_2_ NW−P_2_W_17_V film were also recorded, as shown in Figure 3c–f and Appendix A. Combined with the TEM morphological observations, the distributions of W, P, Ti, and V suggest a uniform distribution of P_2_W_17_V on the TiO_2_ NWs. As shown in the TEM image (Figure 3b), following the LbL assembly process, the P_2_W_17_V coating layer covered the surface of the NWs, forming a core-shell structure. As indicated by the arrows, the darker columnar area is a TiO_2_ NW and the lighter part surrounding it are P_2_W_17_V particles. The selected area electron diffraction pattern showed the specific diffraction spots of TiO_2_ nanowires, and it can be attributed to the rutile phase [23].

### 3.2. EC Performance

To explore the potential of the prepared composite material for application as EC supercapacitor, its EC properties were investigated and compared with those of the FTO−P_2_W_17_V film. As demonstrated in Figure 4a–b, the transmittance was reduced along potentials ranging from 0 to −1.0 V. In addition, as shown in Figure 4c, the maximum transmittance modulation of the NW−P_2_W_17_V film (38.32%) was significantly higher than that of the FTO−P_2_W_17_V film (22.25%) at 580 nm, thereby indicating that the effective combination of two cathodic EC materials could indeed improve the overall performance. For the switching kinetics, the fast switching speed (i.e., the time required to achieve 90% of full modulation) for each of the two prepared films was determined, as shown in Figure 4d. Notably, FTO−P_2_W_17_V (t_c_ = 1.49 s and t_b_ = 1.65 s) and NW−P_2_W_17_V (t_c_ = 1.65 s and t_b_ = 1.64 s) films could undergo relatively rapid colouring and bleaching processes, which are important processes in the context of EC applications. Furthermore, as shown in the optical photograph presented in Figure 4e, the P_2_W_17_V-modified film turned blue, and became deeper in colour upon increasing the applied potential; this colour was attributed to the intervalence charge-transfer band (W^V^–O–W^VI^ or W^VI^–O–W^V^). The transmittance showed a good linear relationship with the applied potential, indicating that the colouration state could be adjusted precisely, thereby rendering this system suitable for practical use in industry. 

The CE is a crucial factor in evaluating the correlation between the change in colour and the number of injected charges. The CE can be calculated from Equations (1) and (2) [31,32,33]:CE = ΔOD/(Q/A)(1)
ΔOD(λ) = log T_b_/T_c_(2)
where Q is the charge density, A is the area of the composite film, and T_b_ and T_c_ are the transmittances of the film in the bleached and coloured states at a certain wavelength (λ), respectively. Figure 4f shows the variation in the optical density with respect to the extent of electric charge exchange from the electrolyte to the EC film. The CE can be obtained from the slope of the line that fits the linear region of the plot. Thus, the CE values of samples were calculated to be 116.5 cm^2^ C^−1^ for NW−P_2_W_17_V and 15.2 cm^2^ C^−1^ for FTO−P_2_W_17_V, wherein the larger value obtained for the NW−P_2_W_17_V system indicates that a large transmittance modulation can be realised through the introduction of a small amount of charge.

The electrochemical stability of a film is vital for determining its EC performance. Thus, the cycling stabilities of the FTO−P_2_W_17_V and NW−P_2_W_17_V films were tested by chronoamperometry at 580 nm over 1000 cycles. As shown in Figure 4g–h, NW−P_2_W_17_V exhibited a superior cycling stability with an initial transmittance variation of approximately 38.32%, wherein ~86% of the initial value was retained after 1000 cycles. This outstanding cycling stability should permit long-term application in real environments.

### 3.3. Energy-Storage Performance

The electrochemical performances of the thin films were then evaluated using CV and galvanostatic charge-discharge (GCD) tests. Figure 5a shows the CV curves of the NW−P_2_W_17_V film measured at different scan rates, wherein it can be seen that upon increasing the scan rate from 50 to 150 mV s^−1^, no obvious changes in shape were observed for the CV curves, although the peak potential moved slightly. The presence of characteristic symmetric reversible peaks for the NW−P_2_W_17_V film also indicate its good capacitive behaviour upon ion insertion/extraction. Furthermore, the inset of Figure 5a shows a good linear relationship between the current density and the scan rate, indicating a fast electron transfer kinetic characteristic in these redox-active materials, which therefore represents a typical surface-controlled process. Figure 5b shows the CV curves of the NW−P_2_W_17_V and FTO−P_2_W_17_V films obtained using a three-electrode system at the same scan rate in a solution of HOAc-NaAc at pH 3.5. The composite film displayed three pairs of redox peaks, which can be attributed to the redox reaction between W^VI^ and W^V^, indicating a typical faradic behaviour. The redox peaks of the NW−P_2_W_17_V film have higher peak current values than those of the FTO−P_2_W_17_V film, indicating the high conductivity and low internal resistance of the NW−P_2_W_17_V film. These increased peak current values can be attributed to the influence of faradaic reactions and to hydrogen ion (H^+^) intercalation at the electrode/electrolyte interface.

The diffusion coefficient of H^+^ ions for insertion and extraction can be estimated based on the measured peak current, I_p_ (A) [34,35]:(3)$$I_{p}=2.69\;\times \;10^{5}\mathrm{AC}\sqrt{Dvn^{3}}$$
where I_p_ is the peak current, A is the area of the film (cm^2^), n is the number of electrons, D is the diffusion coefficient of the H^+^ ions (cm^2^ s^−1^), C is the concentration of the H^+^ ions in the electrolyte solution (mol cm^−3^), and v is the scan rate (V s^−1^). The diffusion rate of H^+^ in NW−P_2_W_17_V was faster than that in FTO−P_2_W_17_V. This enhanced diffusion rate for NW−P_2_W_17_V therefore accounted for the superior electrical conductivity of this material.

Owing to their fast ion intercalation/deintercalation properties and excellent cycling stabilities, we envisaged that the composite films could have great potential for use in energy-storage applications. Thus, to further evaluate the capacitive behaviours of the composite films, a series of GCD measurements were carried out at different current densities. Figure 5c shows the potential responses of the NW−P_2_W_17_V film under different currents, in addition to the dependence of the volumetric capacitance of the composite film on the current density. The GCD curves collected under different current densities are displayed in Figure 5c, which shows that the shapes of the CD profiles were essentially retained for all the applied current ranges, demonstrating the superior charge/discharge reversibility of the sample [36]. Plateau regions are observed in the GCD curves, and the positions of the three plateaus are consistent with the CV curves, thereby indicating that the capacitance is mainly caused by the faradaic redox reaction, whereas the existence of plateaus in the curves illustrates a sound pseudocapacitive behaviour [37,38]. The calculated volumetric capacitance as a function of the current density is shown in Figure 5d. The volumetric capacitance gradually declines as the current density increases, mainly because the limited ion diffusion rate is inaccessible, and so adequate surface redox reactions of the active materials cannot be ensured at high current densities. Furthermore, the value obtained for the NW−P_2_W_17_V film was higher than that of the FTO−P_2_W_17_V film, which was ascribed to the interactions and synergistic effects between the P_2_W_17_V and TiO_2_ NW materials. Furthermore, the GCD curves at 0.3 mA cm^−2^ and the corresponding in situ transmittance at 580 nm were collected and plotted in Figure 5e. During the charging process, the NW−P_2_W_17_V electrode gradually became coloured, and the decrease in transmittance was distinguishable. In contrast, the colour of the electrode was reversibly bleached during the discharge process.

The long-term cycling stability is another vital index for evaluating the properties of electrode materials [39,40]. As shown in Figure 5f, the NW−P_2_W_17_V film revealed an excellent cyclic stability with its volumetric capacitance being almost fully maintained after 1000 cycles at 0.2 mA cm^−2^ in a voltage range of −0.5 to 0.2 V.

Subsequently, electrochemical impedance spectroscopy (EIS) was employed to investigate the inner resistances and capacitance properties of the thin films [30]. Figure 6a shows the Nyquist plots of the NW−P_2_W_17_V and FTO−P_2_W_17_V films with a frequency range of 0.01–100,000 Hz and a signal amplitude of ±5 mV. The electrode system can be described by a simple equivalent circuit (see the inset of Figure 6a), which was selected to fit the obtained impedance data for the NW−P_2_W_17_V composite film. The high-frequency part of the semicircle in the EIS spectrum indicates the speed of the electron transfer process, and the diameter is closely related to the electron transfer resistance (Rct). The Rct of the FTO−P_2_W_17_V film was significantly smaller than that of the NW−P_2_W_17_V film, indicating the lower Rct and the higher electron transfer rate of NW−P_2_W_17_V composite film. As outlined in Figure 6b, we constructed an EES device using LiClO_4_/PC as the electrolyte, the NW−P_2_W_17_V composite film as the negative electrode, and FTO as the positive electrode. Importantly, this EES device was capable of lighting a red LED (Figure 6c). After charging for 10 s, the device became dark blue in colour, and the system lit the red LED for a total of 20 s. These results indicate that the energy-storage states were directly reflected by the colour change. More specifically, as the charge stored inside the device increased, its colour deepened. Overall, these observations verify the potential practical application of our device in energy-storage smart windows and visual monitoring systems.

## 4. Conclusions

In this work, a suitably designed nanocomposite film composed of vanadium-substituted Dawson-type POMs were fabricated on a TiO_2_ nanowire array substrate. Compared with the dense packing structure, the core—shell nano structure exhibited enhanced EC and electrochemical properties with significant optical contrast (38.32% at 580 nm), short response time (1.65 and 1.64 s for colouring and bleaching, respectively), and satisfactory volumetric capacitance (297.1 F cm^−3^ at 0.2 mA cm^−2^), which mainly originate from the unique three-dimensional structure of a nanocomposite with low tortuosity and a high specific surface area. TiO_2_ NW not only provided a transparent substrate with greater adhesion, but it also shortened the electrons/ions diffusion pathway, resulting in uniform and fast reaction kinetic characteristics. A solid-state EES device was fabricated using the composite film as the cathode. In terms of its potential practical applications, the developed device was demonstrated to light up a red LED, and the energy-storage state of the device was easily monitored by observing its change in colour, so as to achieve the purpose of real-time monitoring, and avert the damage caused by overcharging and over-discharging to the supercapacitor. These results therefore confirm the promising features of POM-based EES devices and demonstrate their potential for use in a wide range of multifunctional supercapacitors, such as self-charging supercapacitors, smart energy storage windows, and electrochromic supercapacitors.

## Figures and Tables

**Figure 1 molecules-27-04291-f001:**
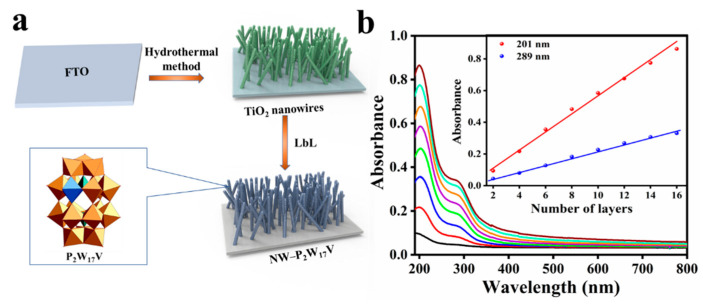
Characterization of material preparation process. (**a**) Schematic outline of the fabrication process of nanocomposite film; (**b**) UV-vis absorption spectra of composite film on quartz substrate (number of cycles: 2–16). Inset: plots of the absorbance values at 201 and 289 nm as a function of the layer number.

**Figure 2 molecules-27-04291-f002:**
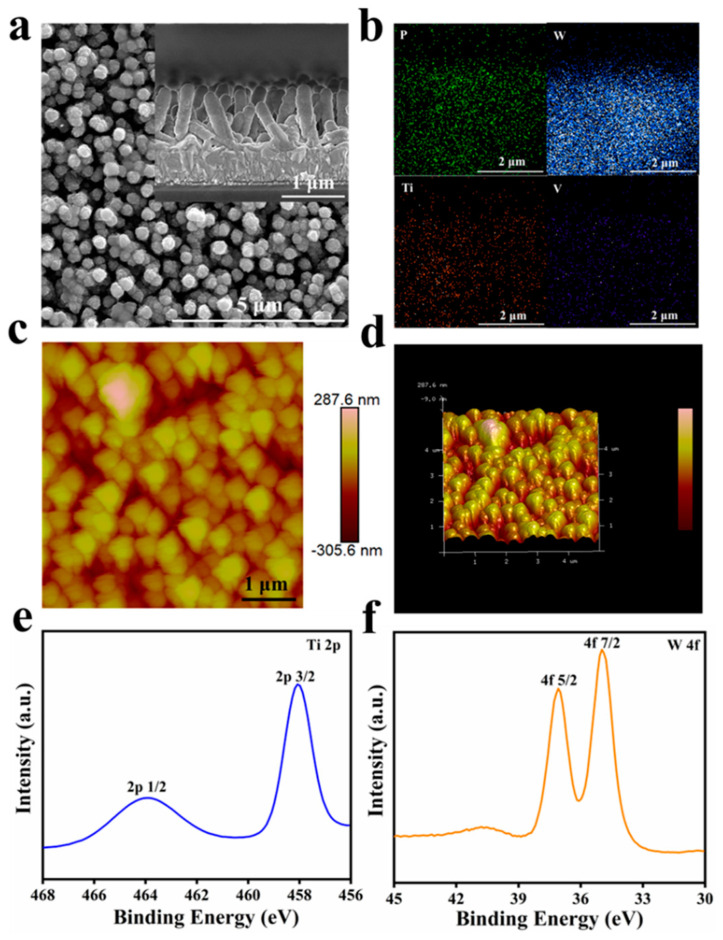
(**a**) SEM images of NW−P_2_W_17_V (inset: the cross-sectional images of prepared films); (**b**) EDS mapping of NW−P_2_W_17_V for P, W, Ti and V respectively; (**c**) 2D AFM images; and (**d**) 3D AFM images of NW−P_2_W_17_V films; High-resolution XPS spectra for Ti 2p (**e**) and W 4f (**f**).

**Figure 3 molecules-27-04291-f003:**
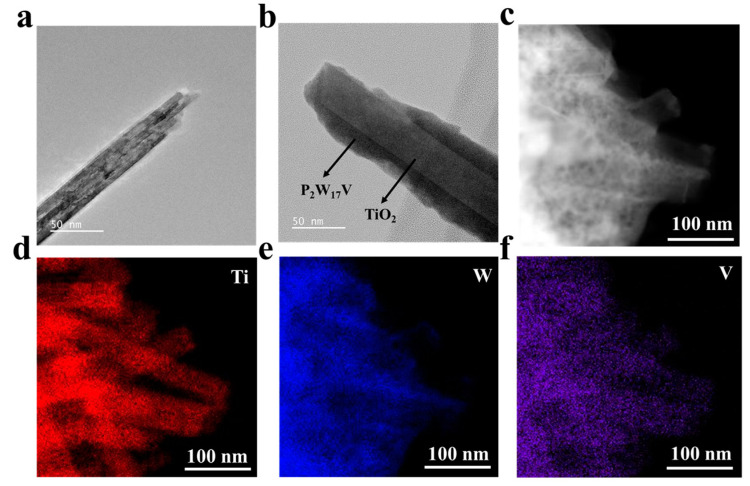
TEM images of TiO_2_ nanowires (**a**) and NW−P_2_W_17_V (**b**); (**c**–**f**) EDS elemental mapping patterns of Ti, W, and V in the NW−P_2_W_17_V films.

**Figure 4 molecules-27-04291-f004:**
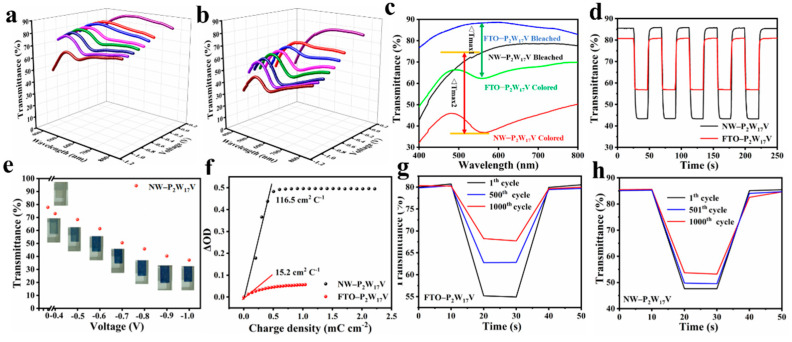
Visible transmittance spectrum of FTO−P_2_W_17_V (**a**) and NW−P_2_W_17_V (**b**) films at different potentials. (**c**) Visible spectra of prepared films at colored and bleached state; (**d**) Chronoamperometry measurements and corresponding in situ optical transmittance curves for FTO−P_2_W_17_V and NW−P_2_W_17_V films at 580 nm; (**e**) Plots of the transmittance value versus applied voltage for NW−P_2_W_17_V and corresponding optical images; (**f**) Coloration efficiency at 580 nm of NW−P_2_W_17_V and FTO−P_2_W_17_V films during subsequent double-potential steps (−1 V and +1 V); Cycle stability of FTO−P_2_W_17_V (**g**) and NW−P_2_W_17_V films (**h**) at 580 nm under square wave potentials of −1 V and +1 V.

**Figure 5 molecules-27-04291-f005:**
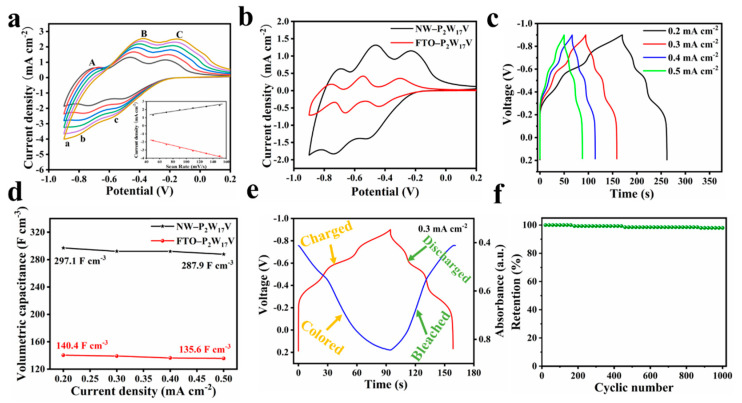
(**a**) CV for the NW−P_2_W_17_V film at different scan rates (from inner to outer): 50, 70, 90, 110, 130, and 150 mV s*^−^*^1^. The inset shows plots of the anodic and the cathodic peak currents for C-c against scan rates; (**b**) CV for NW−P_2_W_17_V and FTO−P_2_W_17_V films at a scan rate of 50 mV/s; (**c**) Charge/discharge curves of NW−P_2_W_17_V film at various current densities; (**d**) Volumetric capacitance at various current densities of NW−P_2_W_17_V and FTO−P_2_W_17_V films; (**e**) In situ transmittance evolution at 580 nm with the charging and discharging process of the NW−P_2_W_17_V film; (**f**) Cycle performance of NW−P_2_W_17_V film measured under a current density of 0.2 mA cm*^−^*^2^.

**Figure 6 molecules-27-04291-f006:**
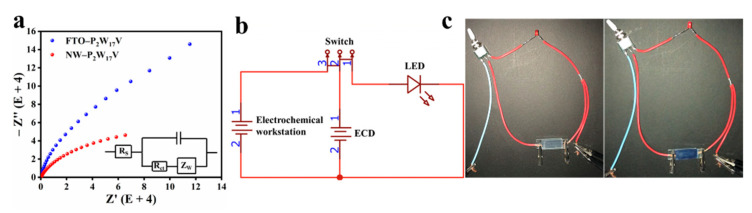
(**a**) The EIS figures of the films with different components FTO−P_2_W_17_V and NW−P_2_W_17_V film, the inset shows a simple equivalent circuit about the NW−P_2_W_17_V electrode system; (**b**) Structural diagram of the solid-state EC device architecture used in this work; (**c**) Photo of a red LED lit up and out by a solid-state EC device.

## Data Availability

Data are contained within the article and Appendix A.

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
