# Peer review of "Vanadium-Substituted Dawson-Type Polyoxometalate–TiO2 Nanowire Composite Film as Advanced Cathode Material for Bifunctional Electrochromic Energy-Storage Devices"

_molecules, 2022, doi:10.3390/molecules27134291_

Round 1
Reviewer 1 Report
The article is well written and brings together some intersting concepts that may be of interest to readers from diverse scientific fields.
A few minor improvements should be made to increase the scientific depth of the article and broaden its appeal.
The density of energy storage should be critically examined with reference to similar materials and studies. Authors should clarify to what category of energy storage devices the proposed systems relate to.
The authors mention that TiO2 nanostructures are useful or promising for energy storage applications. This should be expanded and explained more comprehensively. Why is this oxide used and in which applications? What role does the semiconductor band-gap of tio2 play in its performance?
The phase and growth direction of TiO2 are of considerable importance, and yet authors do not mention which phase of TiO2 is used here, Anatase or Rutile? It is probably Rutile, growing in a favourable orientation, more information should be provided in this regards. Distinguishing between rutile nanowires and nanorods would also be useful. Authors should refer to and cite the articles
https://doi.org/10.1016/j.jcrysgro.2012.08.015
https://doi.org/10.1021/ja8078972
https://doi.org/10.1021/cg500743u
And explain concisely how the growth habit and dimension govern the levels of exposed surfaces present in the material
Author Response
Reviewer 1
The article is well written and brings together some interesting concepts that may be of interest to readers from diverse scientific fields. A few minor improvements should be made to increase the scientific depth of the article and broaden its appeal.
Reply: We appreciate Reviewer #1’s comments very much. The comments really help us to improve the manuscript. We revised the paper and marked it in red.
“As shown in the TEM image (Figure 3b), following the LbL assembly process, the P2W17V coating layer covered the surface of the NWs, forming a core-shell structure. As indicated by the arrows, the darker columnar area is a TiO2 NW and the lighter part surrounding it are P2W17V particles. The selected area electron diffraction pattern showed the specific diffraction spots of TiO2 nanowires, and it can be attributed to the rutile phase [23].” (Page 6 line 222~227)
“Compared with the dense packing structure, the core−shell nano structure exhibited enhanced EC and electrochemical properties with significant optical contrast (38.32% at 580 nm), short response time (1.65 and 1.64 s for colouring and bleaching, respectively), and satisfactory volumetric capacitance (297.1 F cm−3 at 0.2 mA cm−2), which mainly originate from the TiO2 NW backbone with low tortuosity and high specific surface area. TiO2 NW not only provided a transparent substrate with greater adhesion but also shorten the electrons/ions diffusion pathway, resulting in uniform and fast reaction kinetic characteristics.” (Page 10 line 370~377)
Detailed answers are listed as follows:
- The density of energy storage should be critically examined with reference to similar materials and studies. Authors should clarify to what category of energy storage devices the proposed systems relate to.
Reply: The POMs-based bifunctional electrochromic energy-storage material reported herein can realize different colors by applying different voltages, thus achieving the purpose of real-time monitoring of the energy storage level, providing useful references for the development of multifunctional supercapacitors, such as self-charging supercapacitors, smart energy storage windows, and electrochromic supercapacitors. At present, inorganic, organic and inorganic-organic hybrid materials are widely used in electrochromic energy-storage devices. The electrochromic and electrochemical parameters of some similar materials are shown in Table 1, which can be seen that the energy storage and electrochromic properties reported here are relatively good.
Table 1 Electrochromic and electrochemical performance parameters of electrochromic capacitors
|
Number |
Materials |
AC (mF cm-2) or SC (F g-1) or VC (F cm-3) |
Optical modulation (%) |
CE (cm2 C-1) |
tc/tb (s) |
Ref. |
|
1 2 3 4 5 6 7 8 9 10 11 12 13 14 |
Mo-doped WO3 PRF-WO3 NiO-CO PANI-CNT Hybrid WO3 nanoarrays PEDOT-HS AgNW/WO3 AgNWs/rGO/WO3 WO3 NiO/Ag/NiO NW-P2W17-Fe(phen)3 NW/P2W17/Cu(phen)2 NW-P2W17 NW-P2W17V |
19.1/AC 44.0/AC 88.24/AC 336.9/SC 47.4/AC 121.6/SC 13.6/AC 406/SC 639.8/AC 364/SC 135.8/VC 228/VC 172.3/VC 297.1/ VC |
60 - 52 46.6 - 35.6 55.9 83.4 91.3 70 34.3 43.7 33.5 38.32 |
84.3 56.8 - 120 92.3 153.4 80.2 64.8 54.8 76.6 194.5 50.4 150.34 116.5 |
4.9/4.0 3.2/5.6 5.9/7.1 5.8/7.9 3.0/3.6 1.1/1.2 1.7/1.0 - 3.1/0.9 4.3/4.0 2.8/6.2 2.3/9.5 9.05/1.69 1.65/1.64 |
1 2 3 4 5 6 7 8 9 10 11 12 13 Our work |
References
- Xie S. J, Chen Y B, Bi Z J, Jia S S, Guo X X, Gao X D and Li X M. Energy storage smart window with transparent-to-dark electrochromic behavior and improved pseudocapacitive performance. Chem. Eng. J. 2019, 370, 1459–1466.
- Shi Y D, Sun M. J, Chen W J, Zhang Y, Shu X, Qin Y Q, Zhang X R, Shen H J and Wu Y C. Rational construction of porous amorphous WO3 nanostructures with high electrochromic energy storage performance: Effect of temperature. J. Non-Cryst. Solids. 2020, 549, 120337.
- Xue J Y, Li W J, Song Y, Li Y and Zhao J P. Visualization electrochromic-supercapacitor device based on porous Co doped NiO films. J. Alloys Compd. 2021, 857, 158087.
- Xu K, Zhang Q Q, Hao Z D, Tang Y H, Wang H, Liu J B and Yan H. Integrated electrochromic supercapacitors with visual energy levels boosted by coating onto carbon nanotube conductive networks. Sol. Energy Mater. Sol. Cells. 2020, 206, 110330.
- Shi Y D, Sun M J, Zhang Y, Cui J W, Wu Y Q, Ta H H, Liu J Q and Wu Y C. Structure modulated amorphous/crystalline WO3 nanoporous arrays with superior electrochromic energy storage performance. Sol. Energy Mater. Sol. Cells. 2020, 212, 110579.
- Zhang S H, Ren J Y, Zhang Y, Peng H C and Chen S. PEDOT hollow nanospheres for integrated bifunctional electrochromic supercapacitors. Organic Electronics, 2019, 77, 105497.
- Shen L X, Du L H, Tan S Z, Zang Z G, Zhao C. X and Mai W J. Flexible electrochromic supercapacitor hybrid electrodes based on tungsten oxide films and silver nanowires, Chem. Commun. 2016, 52, 6296–6299.
- Yun T G, Kim D H, Kim Y H, Park M K, Hyun S M and Han S M. Photoresponsive smart coloration electrochromic supercapacitor. Adv. Mater. 2017, 29, 1606728.
- Yang P H, Sun P, Chai Z S, Huang L H, Cai X, Tan S Z, Song J H and Mai W J. Large-scale fabrication of pseudocapacitive glass windows that combine electrochromism and energy storage. Angew. Chem. Int. Ed. 2014, 53, 11935–11939.
- Dong W J, Lv Y, Zhang N, Xiao L L, Fan Y and Liu X Y. Trifunctional NiO–Ag–NiO electrodes for ITO-free electrochromic supercapacitors. J. Mater. Chem. C. 2017, 5, 8408–8414.
- Chu D X, Qu X S, Zhang S F, Zhang J R, Yang Y Y and An W J. Polyoxotungstate-based nanocomposite films with multi-color change and high volumetric capacitance toward electrochromic energy-storage applications. New J. Chem. 2021, 45, 19977–19985.
- Chu D X, Qu X S, Zhang S F, Zhang J R, Liu Z F, Zhou L L and Yang Y Y. Copper complex/polyoxometalate-based tunable multi-color film for energy storage. Asia-Pac J. Chem. Eng. 2022, e2779.
- Qu X S, Fu Y, Ma C, Yang Y Y, Shi D, Chu D X and Yu X Y. Bifunctional electrochromic-energy storage materials with enhanced performance obtained by hybridizing TiO2 nanowires with POMs. New. J. Chem. 2020, 44, 15475–15482.
- The authors mention that TiO2 nanostructures are useful or promising for energy storage applications. This should be expanded and explained more comprehensively. Why is this oxide used and in which applications? What role does the semiconductor band-gap of tio2 play in its performance?
Reply: When choosing materials for energy application, one of the most important aspects is to look for large active surface area and short diffusion path lengths for charge carriers, which are fundamental criteria mainly for the purpose of efficient electrochemical energy storage. An extremely high active surface area can be achieved through the construction of core−shell nano structure, and various types of core−shell nanowire arrays whose core or shell materials consist of inorganic transition metal oxides, such as TiO2[14-19].
In our work, the TiO2 NW substrate could not only act as a framework with a high specific surface area during the redox reaction process but also shorten the ion diffusion pathway and promote the transfer of electrons/ions, resulting in a uniform and fast reaction kinetics characteristic. First, TiO2 NW not only provided a transparent substrate with greater adhesion but also transferred electrons/ions to polyanions under an applied potential; second, the 1D matrix could enhance the specific surface area of electrochromic films and thus increase the amount of active material during the reaction process; furthermore, the EC films prepared via a traditional method could induce the accumulation of active materials, which could lead to high tortuosity and long reaction path, resulting in slow reaction kinetics; however, the TiO2 NW array could effectively solve these problems. On the one hand, the substrate with a unique 3D structure could improve the penetration and diffusion of an electrolyte. On the other hand, another important design advantage of the TiO2 NW matrix is that a shorter ion-diffusion pathway was fabricated, resulting in uniform and fast reaction kinetic characteristics.
As a pseudocapacitive cathode material, a pair of redox peaks appear on the substrate TiO2 at negative voltage, which belongs to the reaction of Ti4+ reduction and H+, Li+, Na+ insertion. The mechanism of color change and energy storage is as follows:
TiO2 + (e- + H+) ⇌ HTiO2
The introduction of TiO2 can synergize with POMs to improve the ion transport rate and capacity performance. However, how the band gap of semiconductor materials affects the energy storage performance remains unclear.
References
- Feng X J, Shankar K, Varghese O K, Pauloe M, Latempa T J and Grimes C A. Vertically aligned single crystal TiO2 nanowire arrays grown directly on transparent conducting oxide coated glass: Synthesis details and applications. Nano Lett. 2008, 8, 3781−
- Xia X H, Tu J P, Zhang Y Q, Chen J, Wang X L, Gu C D, Guan C, Luo J S and Fan H J. Porous hydroxide nanosheets on preformed nanowires by electrodeposition: Branched nanoarrays for electrochemical energy storage. Chem. Mater. 2012, 24, 3793−3799.
- Cai G F, Tu J P, Zhou D, Li L, Zhang J H, Wang X L and Gu C D. Constructed TiO2/NiO core/shell nanorod array for efficient electrochromic application. J. Phys. Chem. C. 2014, 118, 6690−6696.
- Barawi M, Trizio L D, Giannuzzi R, Veramonti G, Manna L and Manca M. Dual band electrochromic devices based on Nb-Doped TiO2 nanocrystalline electrodes. ACS Nano. 2017, 11, 3576−3584.
- Mishra S, Yogi P, Sagdeo P R and Kumar R. TiO2−Co3O4core−shell nanorods: bifunctional role in better energy storage and electrochromism. ACS Appl. Energy Mater. 2018, 1, 790−798
- Truong Q D, Le T S and Hoa T H. Ultrathin TiO2 rutile nanowires enable reversible Mg-ion intercalation. Mater. Lett. 2019, 254, 357–360.
- The phase and growth direction of TiO2 are of considerable importance, and yet authors do not mention which phase of TiO2 is used here, Anatase or Rutile? It is probably Rutile, growing in a favourable orientation, more information should be provided in this regards. Distinguishing between rutile nanowires and nanorods would also be useful. Authors should refer to and cite the articles
Reply: The phase of TiO2 nanowires has been investigated in our previous paper[20]. The SAED image (Figure 1) showed the specific diffraction spots of TiO2 nanowires and the presented diffraction rings could be attributed to (101) and (002) planes of rutile phase TiO2 respectively.
Figure 1. SAED pattern images of the prepared TiO2 nanowires.
Nanorods generally refer to cylindrical (or polyangular cross-section) solid nanomaterials with shorter length and straight longitudinal shape[21-23]; nanowires are solid nanomaterials with longer length and straight or curved morphology[24, 25]. However, the definition and distinction of nanorods and nanowires are relatively vague. The ratio of the length and diameter of the TiO2 nanostructures fabricated here is relatively large, so we identified them as nanowires.
References
- Qu X S; Ma C, Fu Y, Liu S P, Wang J and Yang Y Y. Construction of a vertically arrayed three-dimensional composite structure as a high coloration efficiency electrochromic film. New. J. Chem. 2020, 44, 4177-−
- Qi W Q, Du J, Peng Y C, Wu W H, Zhang Z J, Li X Y, Li K, Zhang K, Gong C, Luo M and Peng H L. Hydrothermal synthesis of TiO2 nanorods arrays on ITO. Mater. Chem. Phys. 2018, 207, 435−
- Mishra S, Yogi P, Sagdeo P R and Kumar R. TiO2–Co3O4 core–shell nanorods: bifunctional role in better energy storage and electrochromism. ACS Appl. Energy Mater. 2018, 1, 790−
- Cai G F, Tu J P, Zhou D, Li L, Zhang J H, Wang X L and Gu C D. Constructed TiO2/NiO core/shell nanorod array for efficient electrochromic application. J. Phys. Chem. C 2014, 118, 6690−6696.
- He J J, Wang M, Wu X F, Sun Y, Huang K K, Chen H W and Gao L. Influence of controlled Pd nanoparticles decorated TiO2 nanowire arrays for efficient photoelectrochemical water splitting. Alloys Compd. 2019, 785, 391−397.
- Liu S P, Zhang X T, Sun P P, Wang C H, Wei Y G and Liu Y C. Enhanced electrochromic properties of a TiO2 nanowire array via decoration with anatase nanoparticles. J. Mater. Chem. C. 2014, 2, 7891−
Reviewer 2 Report
The manuscript reports on fabrication and study of a new nanocomposite material composed of vanadium-containing phosphotungstate and titanium dioxide nanowire. The new material was probed in practical terms for application in the field of electrochromic energy-storage devices.
The manuscript is well-written, the findings are supported by the data. I suppose it can be published in current form. I would advise only to develop the Conclusions since they are too brief and do not fully reflect the content of the article.
Author Response
Reviewer 2
The manuscript reports on fabrication and study of a new nanocomposite material composed of vanadium-containing phosphotungstate and titanium dioxide nanowire. The new material was probed in practical terms for application in the field of electrochromic energy-storage devices.
The manuscript is well-written, the findings are supported by the data. I suppose it can be published in current form. I would advise only to develop the Conclusions since they are too brief and do not fully reflect the content of the article.
Reply: Thanks so much for your careful review and valuable suggestions. Your suggestions are very helpful for us to reinforce our manuscript. The initial conclusions has been revised carefully according to your comments. We revised the paper and marked it in red.
“In this work, a suitably designed nanocomposite film composed of vanadium-substituted Dawson-type POMs were fabricated on a TiO2 nanowire array substrate. Compared with the dense packing structure, the core−shell nano structure exhibited enhanced EC and electrochemical properties with significant optical contrast (38.32% at 580 nm), short response time (1.65 and 1.64 s for colouring and bleaching, respectively), and satisfactory volumetric capacitance (297.1 F cm−3 at 0.2 mA cm−2), which mainly originate from the unique three-dimensional structure of nanocomposite with low tortuosity and high specific surface area. TiO2 NW not only provided a transparent substrate with greater adhesion but also shorten the electrons/ions diffusion pathway, resulting in uniform and fast reaction kinetic characteristics. A solid-state EES device was fabricated using the composite film as the cathode. In terms of its potential practical applications, the developed device was demonstrated to light up a red LED, and the energy-storage state of the device was easily monitored by observing its change in colour, so as to achieve the purpose of real-time monitoring, and avert the damage caused by overcharging and over-discharging to the supercapacitor. These results therefore confirm the promising features of POM-based EES devices and demonstrate their potential for use in a wide range of multifunctional supercapacitors, such as self-charging supercapacitors, smart energy storage windows, and electrochromic supercapacitors.” (Page 10 line 368~385)